# CONFIDENCE SCORES MAKE INSTANCE-DEPENDENT LABEL-NOISE LEARNING POSSIBLE

## ABSTRACT

Learning with noisy labels has drawn a lot of attention. In this area, most of recent works only consider *class-conditional noise*, where the label noise is independent of its input features. This noise model may not be faithful to many real-world applications. Instead, few pioneer works have studied *instance-dependent noise*, but these methods are limited to strong assumptions on noise models. To alleviate this issue, we introduce *confidence-scored instance-dependent noise* (CSIDN), where each instance-label pair is associated with a *confidence score*. The confidence scores are sufficient to estimate the noise functions of each instance with minimal assumptions. Moreover, such scores can be easily and cheaply derived during the construction of the dataset through crowdsourcing or automatic annotation. To handle CSIDN, we design a benchmark algorithm termed *instance-level forward correction*. Empirical results on synthetic and real-world datasets demonstrate the utility of our proposed method.

## 1 INTRODUCTION

The recent success of deep neural networks has increased the need for high-quality labeled data. However, such a labelling process can be time-consuming and costly. A compromise is to resort to weakly-supervised annotations, using crowdsourcing platforms or trained classifiers that annotate the data automatically. These weakly-supervised annotations tend to be low-quality and noisy, which negatively affects the accuracy of high-capacity models due to memorization effects (Zhang et al., 2017). Thus, learning with noisy labels has often drawn a lot of attention.

Early works on noisy labels studied *random classification noise* (RCN) for binary classification (Angluin & Laird, 1988; Kearns, 1993). In the RCN model, each instance has its label flipped with a fixed noise rate $\rho \in [0, \frac{1}{2})$. A natural extension of RCN is *class-conditional noise* (CCN) for multi-class classification (Stempfel & Ralaivola, 2009; Natarajan et al., 2013; Scott et al., 2013; Menon et al., 2015; van Rooyen & Williamson, 2015; Patrini et al., 2016) (Appendix A). In the CCN model, each instance from class $i$ has a fixed probability $\rho_{i,j}$ of being assigned to class $j$. Thus, it is possible to encode some similarity information between classes. For example, we can expect that the image of a "dog" is more likely to be erroneously labelled as "cat" than "boat".

To handle the CCN model, a common method is the *loss correction*, which aims to correct the prediction or the loss of the classifier using an estimated noise transition matrix (Patrini et al., 2017; Sukhbaatar et al., 2015; Goldberger & Ben-Reuven, 2017; Ma et al., 2018). Another common approach is the *label correction*, which aims to improve the label quality during training. For example, Reed et al. (2015) introduced a bootstrapping scheme. Similarly, Tanaka et al. (2018) proposed to update the weights of a classifier iteratively using noisy labels, and use the updated classifier to yield more high-quality pseudo-labels for the training set. Although these methods have theoretical guarantees, they are unable to cope with real-world noise, e.g., *instance-dependent noise* (IDN).

The IDN model considers a more general noise (Manwani & Sastry, 2013; Ghosh et al., 2014; Menon et al., 2016; Cheng et al., 2017; Menon et al., 2018), where the probability that an instance is mislabeled depends on both its class and features. Intuitively, this noise is quite realistic, as poor-quality or ambiguous instances are more likely to be mislabeled in real-world datasets. However, it is much more complex to formulate the IDN model, since the probability of a mislabeled instance is a function of not only the label space but also the input space that can be very high dimensional.

Table 1: Comparisons between baselines and our work for handling the IDN model. Rate identifiability denotes whether the transition matrix is identifiable.

| Approaches | Multi-class | Rate-identifiability | Unbounded-noise |
|---|---|---|---|
| Du & Cai (2015) | ✗ | ✗ | ✓ |
| Menon et al. (2018) | ✗ | ✓ | ✓ |
| Bootkrajang & Chaijaruwanich (2018) | ✗ | ✗ | ✓ |
| Cheng et al. (2017) | ✗ | ✓ | ✗ |
| Our work | ✓ | ✓ | ✓ |

As a result, several pioneer works have considered stronger assumptions on noise functions. However, stronger assumptions tend to restrict the utility of these works (Table 1). For instance, the boundary-consistent noise model considers stronger noise for samples closer to the decision boundary of the Bayesian optimal classifier (Du & Cai, 2015; Menon et al., 2018). However, such a model is restricted to binary and cannot estimate noise functions. Cheng et al. (2017) recently studied a particular case of the IDN model, where noise functions are upper-bounded. Nonetheless, their method is limited to binary classification and has only been tested on small datasets.

Instead of simplifying assumptions on noise functions, we propose to tackle the IDN model from the source, by considering *confidence scores* to be available for the label of each instance. We term this new setting *confidence-scored instance-dependent noise* (CSIDN, Figure 1c). The confidence scores denote how likely an instance is to be correctly labeled. Assuming that (i) confidence scores are available for each instance, (ii) transitions probabilities to other classes are independent of the instance conditionally on the assigned label being erroneous and (iii) a set of anchor points is available, we derive an *instance-level forward correction* algorithm which can fully estimate the transition probability for each instance, and subsequently train a robust classifier with a loss-correction method similarly to Patrini et al. (2017).

It is noted that confidence scores can be easily and cheaply derived during the construction of the dataset. For example, in crowdsourcing platforms, simply counting how many annotators agree on a given instance can give a notion of how confident a label is. Besides, many real-world datasets are automatically annotated using a trained classifier, such as web-scraped datasets (Tong Xiao et al., 2015) and physiological features inferred from medical records (Agarwal et al., 2016). In these cases, the class-probabilities of the labels assigned by the classifier can be seen as confidence scores, provided that the classifier is well calibrated (Guo et al., 2017).

To sum up, we first formulate *instance-dependent noise* in Section 2.1, and expose its robustness challenge in Section 2.2. Then, we explain our motivation to use confidence scores, and propose the *confidence-scored instance-dependent noise* (CSIDN) model in Section 2.3. Lastly, to handle this new noise model, we present the first practical algorithm termed *instance-level forward correction* in Section 3, and validate the proposed algorithm through extensive experiments in Section 4.

## 2 TACKLING INSTANCE-DEPENDENT NOISE FROM THE SOURCE

In this section, we present the IDN model along with the limitations of existing approaches, and introduce the CSIDN model as a tractable instance-dependent noise model.

### 2.1 NOISE MODELS: FROM CLASS-CONDITIONAL TO INSTANCE-DEPENDENT NOISE

We formulate the problem of learning with noisy labels in this section. Let $D$ be the distribution of a pair of random variables $(X, Y) \in \mathcal{X} \times \mathcal{Y}$, where $\mathcal{X} \in \mathbb{R}^d$, $\mathcal{Y} = \{1, 2, \ldots, K\}$ and $K$ is the number of classes. In the classification task with noisy labels, we hope to train a classifier while having only access to samples from a noisy distribution $\bar{D}$ of random variables $(X, \bar{Y}) \in \mathcal{X} \times \mathcal{Y}$. Given a point $x$ sampled from $X$, $\bar{Y}$ is derived from the random variable $Y$ via a noise transition matrix $T(x) = (T_{i,j}(x))_{i,j=1}^K \in [0,1]^{K \times K}$:

$$\forall 1 \leq j \leq K, \ P(\bar{Y} = j | X = x) = \sum_{i=1}^K T_{i,j}(x) P(Y = i | X = x). \tag{1}$$

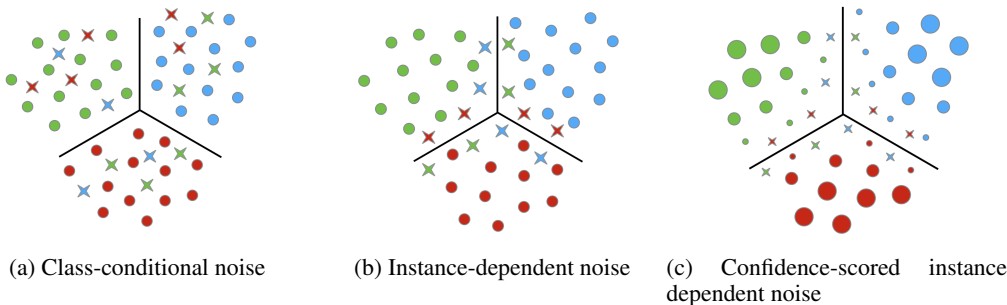

(a) Class-conditional noise     (b) Instance-dependent noise     (c) Confidence-scored instance-dependent noise

Figure 1: Illustration of different noise models. Each color represents an observed class $\bar{y}$: circles indicate $\bar{y} = y$, while crosses indicate $\bar{y} \neq y$. The size of each point represents the confidence scores in the label $\bar{y}$: the bigger the point is, the more confident it is. In the CCN model, the noise function only depends on the label of each instance. In the IDN and CSIDN models, the noise function depends on the observed instance $x$. To illustrate the IDN model, we show a special case called boundary-consistent noise, i.e., points that lie close to the decision boundary are more likely to be mislabelled. Note the CSIDN model varies from the IDN model via confidence scores (Section 2.3).

Each noise function $T_{i,j} : \mathcal{X} \mapsto [0,1]$ is defined as $T_{i,j}(x) = P(\bar{Y} = j | Y = i, X = x)$. In the *class-conditional noise* (CNN) model (Figure 1a), the transition matrix does not depend on the instance $x$ and the noise is entirely characterized by the $K^2$ constants $T_{i,j}$. However, in the *instance-dependent noise* (IDN) model (Figure 1b), the transition matrix depends on the actual instance. This tremendously complicates the problem, as the noise is now characterized by $K^2$ functions over the latent space $\mathcal{X}$, which can be very high dimensional (e.g., $d \sim 10^4$-$10^6$ for an object recognition dataset).

## 2.2 CHALLENGES FROM INSTANCE-DEPENDENT NOISE

**Limitation of existing CCN methods.** Due to the complexity of the IDN model, most recent works in learning with noisy labels have focused on the CCN model (Figure 1a), and the CCN model can be seen as a simplified IDN model (Figure 1b) free of feature information.

In addition to *loss correction* and *label correction* mentioned before, another method for the CCN model is *sample selection*, which aims to find reliable samples during training, such as the small-loss approaches (Jiang et al., 2018; Han et al., 2018). Inspired by the memorization in deep learning (Arpit et al., 2017), those methods first run a standard classifier on a noisy dataset, then select the small-loss samples for reliable training.

However, all approaches cannot handle the IDN model directly. Specifically, *loss correction* considers the noise model to be characterized by a *fixed* transition matrix, which does not include any instance-level information. Meanwhile, *label correction* is vulnerable to the IDN model, since the classifier will be much weaker on noisy regions and labels corrected by the current prediction would likely be erroneous. Similarly, *sample selection* is easily affected by the IDN model.

For example, in the small-loss approaches, instance-dependent noise functions can leave partial regions of the input space clean and other regions very noisy (e.g., in an object recognition dataset, poor-quality pictures will tend to receive more noisy labels than high-quality ones). Since clean regions will tend to receive smaller losses than noisy regions, the small-loss approaches, which only trains on points with the smallest-losses, will focus on clean regions and neglect harder noisy regions. Then, since the distribution of clean regions will subsequently be different from the global distribution, this will introduce a covariate-shift (Shimodaira, 2000), which greatly degrades performances. Moreover, it is hard to use importance reweighting (Sugiyama et al., 2007) for alleviate the issue, since importance reweighting would require estimating the clean posterior probability that is intractable for the IDN model.

To validate this fact, we generate a 3-class distribution of concentric circles (*cf.* Figure 2a), with $\forall (x,y) \in \mathbb{R}^2 \times \{1,2,3\}$, $P(\bar{y} \neq y | x) = \frac{1}{2} \left( \frac{w \cdot x}{\|w\| \|x\|} + 1 \right)$ with $w = (0,1)$ (*cf.* Figure 2b). We

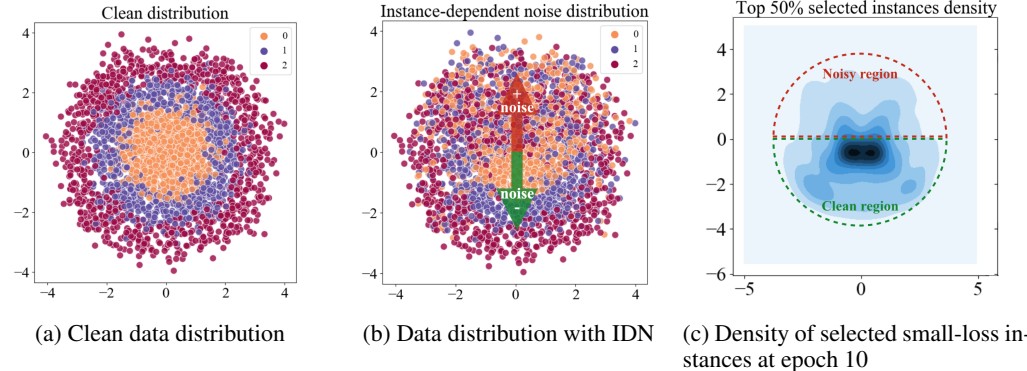

(a) Clean data distribution  (b) Data distribution with IDN  (c) Density of selected small-loss instances at epoch 10

Figure 2: The limitation of the small-loss approaches in the IDN model. (a) Clean distribution. (b) instance-dependent noise in the direction $w = (0, 1)$ with an average corruption rate of 40%: points towards the upper region are more likely to be corrupted than points towards the bottom region. (c) Density map of the instances selected by a small-loss approach at epoch 10. The sample selection gets biased towards clean regions. Since the clean and noisy regions have different distributions, selecting most instances from clean regions creates a covariate-shift between the training and test distributions, which can greatly degrades performances.

then train a network on the top $R(T)$ small-loss instances at each epoch $T$ based on the losses of the previous epoch, with $R(T)$ decreasing in $T$ as described in Han et al. (2018). Figure 2c shows the density of the top $50\%$ small-loss instances selected after 10 epochs: since noisy regions are associated to higher losses, the network eventually tends to select instances from the clean region and neglect the noisy region. This leads to covariate-shift, which is associated with decreased performances (Shimodaira, 2000).

**Limitation of pioneer IDN methods.** The main challenge of the IDN model is the wide range of possible noise functions included in its formulation. Since each $T_{i,j}(\cdot)$ is a function of the high-dimensional input space $\mathcal{X}$, it is challenging for a model to be flexible enough to fit any real-world noise function while being trainable on corrupted datasets, let alone derive theoretical results. Instead, various recent works have considered stronger assumptions on noise functions.

For instance, *boundary-consistent noise* (BCN), first introduced by (Du & Cai, 2015) and generalized in Menon et al. (2018), considers stronger noise for samples closer to the decision boundary of the Bayesian optimal classifier. This is a reasonable model for noise from human annotators, since "harder" instances (i.e., instances closer to the decision boundary) are more likely to be corrupted. Moreover, it is simple enough to derive some theoretical guarantees, as done in Menon et al. (2018). Additionally, an extension of the BCN model was studied in Bootkrajang & Chaijaruwanich (2018), where the noise function is a Gaussian mixture of the distance to the Bayesian optimal boundary. However, the BCN model and its extension are restricted to binary classification, and their geometry-based assumption becomes difficult to fathom for high-dimensional input spaces.

Furthermore, Cheng et al. (2017) recently studied a particular case of the IDN model, where the probabilities that the true labels of samples flip into corrupted ones have upper bounds. They proposed a method based on distilled samples, where noisy labels agree with the optimal Bayesian classifier on the clean distribution. However, their method is limited to binary classification and has only been tested on small UCI datasets. Table 1 summarizes the characteristics of those approaches.

## 2.3 CONFIDENCE-SCORED INSTANCE-DEPENDENT NOISE

Instead of simplifying assumptions on noise functions, we propose to tackle the IDN model from the source. Namely, we consider that, for each instance, we have access to a measure of confidence in the assigned label. As most of noisy datasets arise from crowdsourcing or automatic annotation, such confidence scores can be easily derived during the dataset construction, often with no extra cost. This allows for a good approximation of noise functions with weaker assumptions.

Before introducing our proposed noise model *confidence-scored instance-dependent noise* (CSIDN, Figure 1c), we first define what are the confidence scores, and explain why the confidence scores are available in real-world applications.

**Definition of confidence scores.** For any data point $(x, \bar{y})$ sampled from the joint distribution $(X, \bar{Y})$, we define the confidence score $r_x$ as follows.

$$r_x = P(Y = \bar{y} | \bar{Y} = \bar{y}, X = x). \tag{2}$$

Namely, the probability that the assigned label is correct.

**Availability of confidence scores.** Our rationale is that in tasks involving instance-dependent noise, the confidence information can be easily derived with no extra cost.

Firstly, in crowdsourcing platforms, when multiple workers manually annotate datasets, an aggregation step is often took to aggregate answers of different workers for each instance (e.g., majority vote). An estimation of $r_x$ could then be derived by taking the ratio of votes for the assigned label on the total number of workers. Moreover, since this estimation would of course be less reliable as the number of workers decreases, an alternative could be to directly ask workers for self-reported confidence scores of their responses (Cosmides & Tooby, 1996; Oyama et al., 2013).

Secondly, the confidence information can also be available in automatic annotation via a softmax output layer of deep neural networks. This layer outputs an estimation of the probability that each class is the true label: when a model outputs a given class with probability 0.9, we expect the predicted class to be true 9 times out of 10 on average. A model that estimates the accurate probability is well-calibrated. Therefore, in the case of labels generated by a well-calibrated model, the softmax probability of the assigned label can be directly interpreted as a confidence measure that the label is correct. Even though Guo et al. (2017) showed that recent deep neural networks are not usually well-calibrated (whereas early shallower networks were, as shown in Niculescu-Mizil & Caruana (2005)), model calibration can be achieved in a relatively straightforward way at the validation time, e.g., using temperature scaling (Section 4.2 in Guo et al. (2017)).

**CSIDN: a tractable instance-dependent noise model.** Recall the intrinsic difficulty of the IDN model: to fully characterize this noise, one would need to estimate $K^2$ functions $T_{i,j}(\cdot)$ over the input space $\mathcal{X}$. This is of course intractable with a finite noisy dataset. This is why pioneer solutions to the IDN model have been so far limited by very strong assumptions.

However, considering additional confidence scores, one can wonder whether such information would make the IDN model tractable with less restrictive assumptions. Hence, we introduce a new and tractable instance-dependent noise model: *confidence-scored instance-dependent noise* (CSIDN, Figure 1c). In this noise model, the training data takes the form $S := \{(x_i, \bar{y}_i, r_{x_i}), i = 1, \ldots, N\}$, where $\{(x_i, \bar{y}_i)\}_i \overset{\text{i.i.d.}}{\sim} \bar{D}$ and $r_{x_i} = P(Y = \bar{y}_i | \bar{Y} = \bar{y}_i, X = x_i)$ is the previously defined confidence scores in the assigned label of a given instance (Eq. (2)). The confidence information $r_x$ is decisive for robustness to instance-dependent noise, as it provides a proxy for the noise functions $T_{i,j}$ of the training data that are often intractable otherwise.

## 3 BENCHMARK SOLUTION FOR HANDLING THE CSIDN MODEL

To tackle the CSIDN model, we propose a benchmark solution. Inspired by *forward correction* (Patrini et al., 2017) for the CCN model, we want to correct each prediction $P(\bar{y}|x)$ with the noise transition matrix $T(x)$. However, the transition matrix for the CSIDN model is instance-dependent, and has to be estimated for each instance $x$. We term our solution *instance-level forward correction*.

### 3.1 ESTIMATING INSTANCE-DEPENDENT TRANSITION MATRIX

Using the confidence scores, we will first estimate the diagonal terms $(T_{i,i}(\cdot))_{i=1}^{K}$ of the transition matrix, and then estimate the non-diagonal ones.

**Diagonal terms.** The diagonal terms of the transition matrix correspond to the probabilities that assigned labels are equal to true labels. However, the confidence scores available are only relevant

to the class corresponding to the observed label. Therefore, we need to proceed differently whether the confidence scores are available for the considered class or not.

First, note that for each sample $(x, \bar{y}, r_x) \in S_i := \{(x, \bar{y}, r_x) \in S | \bar{y} = i\}$, $T_{i,i}(x)$ can be derived for the most part from the confidence scores alone:

$$
\begin{aligned}
\forall (x, \bar{y}, r_x) \in S_i, \ T_{i,i}(x) &= P(\bar{Y} = i | Y = i, X = x) \\
&= P(Y = i | \bar{Y} = i, X = x) \frac{P(\bar{Y} = i | X = x)}{P(Y = i | X = x)} \\
&= r_x \, \beta_i(x),
\end{aligned}
\tag{3}
$$

where $\beta_i(x) = \frac{P(\bar{Y}=i|X=x)}{P(Y=i|X=x)}$.

In practice, we use an iterative procedure to estimate in turn $\beta_i(\cdot)$ and $T_{i,i}(\cdot)$ (see Section 3.2 for details). Then, for the rest of samples $(x, \bar{y}, r_x) \in S \backslash S_i$, $r_x$ does not give any direct information on $T_{i,i}(\cdot)$. Hence, we simply set each function $T_{i,i}(\cdot)$ as its empirical mean $\mu_i$ estimated using samples from $S_i$ at the current epoch:

$$
\forall (x, y, r_x) \in S \backslash S_i, \ \hat{T}_{i,i}(x) = \frac{1}{|S_i|} \sum_{(x', \bar{y}', r'_x) \in S_i} T_{i,i}(x') = \mu_i,
\tag{4}
$$

where $|S|$ denotes the cardinality of $S$.

**Non-diagonal terms.** For non-diagonal terms, we have:

$$
\begin{aligned}
\forall i \neq j, \forall x \in \mathcal{X}, \ T_{i,j}(x) &= P(\bar{Y} = j | Y = i, X = x) \\
&= P(\bar{Y} = j, \bar{Y} \neq i | Y = i, X = x) \\
&= P(\bar{Y} = j | \bar{Y} \neq i, Y = i, X = x) P(\bar{Y} \neq i | Y = i, X = x) \\
&= \alpha_{i,j}(x)(1 - T_{i,i}(x)),
\end{aligned}
\tag{5}
$$

where $\alpha_{i,j}(x) = P(\bar{Y} = j | \bar{Y} \neq i, Y = i, X = x)$.

In Eq. (4), $\alpha_{i,j}(x)$ refers to the probability that an instance $x$ with true label $i$ has an observed label $j$, once we know that the observed label is different from the true one. Then, a reasonable assumption is that $\forall i \neq j, \forall x \in \mathcal{X}, \alpha_{i,j}(x) = \alpha_{i,j}$: conditionally on the observed label being erroneous, the class transitions are not influenced by the instance $x$. In other words, the dependence in $x$ of the noise function only impacts the "magnitude" of the noise and not the class transitions.

To illustrate this assumption, consider a crowdsourcing task of object recognition with adjacent classes which annotators can only differentiate with details that can be more or less visible depending on the instance. For example, objects from a given class may have distinctive traits, but those can be more or less visible in the pictures. When those traits are present, the annotators can confidently predict the right class. Otherwise, they will make errors towards adjacent classes. In this case, the probability that the assigned label is wrong highly depends the instance (with distinctive traits being visible or not). Nonetheless, conditionally on the instance being corrupted, i.e., because those traits were not visible enough on the image, the transition probabilities to the adjacent classes are not influenced by the instance itself.

With the previous assumption, we obtain $\forall i \neq j, \forall x \in \mathcal{X}, T_{i,j}(x) = \alpha_{i,j}(1 - T_{i,i}(x))$ with $\alpha_{i,j} \in [0, 1]$. This allows us to estimate the $K(K-1)$ constants $(\alpha_{i,j})_{i \neq j}$ once, and derive the non-diagonal noise functions of $T(x)$ directly from our estimates of the diagonal noise functions (Eq. (5)).

### 3.2 Overall algorithm: Instance-level forward correction

**Estimating $T_{i,i}$ and $\beta_i$.** To train a classifier $h$ with the *instance-level forward correction* method, we need to estimate both $T_{i,i}(x)$ and $\beta_i(x) = \frac{P(\bar{Y}=i|X=x)}{P(Y=i|X=x)}$ from Eq. (3), for all $x \in S_i$. Firstly, the noisy posterior $P(\bar{Y} = i | X = x)$ can be easily estimated by training a naive classifier on the noisy dataset. Secondly, the true posterior $P(Y = i | X = x)$ can be estimated using the output of the classifier $h(x) = \hat{P}(Y = i | X = x)$ at the previous epoch.

Therefore, we iteratively update $\hat{\beta}$ and $\hat{T}$ with the following steps: 1) $\forall x \in \mathcal{X}$, initialize $\hat{\beta}_i(x) = 1$ and train a naive classifier $h_{\text{noisy}}$ on the noisy data $\bar{D}$ to obtain $h_{\text{noisy}}(x) = \hat{P}(\bar{Y}|X = x)$. 2) $\forall i \in [1, K]$, for each sample $(x, \bar{y}, r_x) \in S_i$, compute $\hat{T}_{i,i}(x) = r_x \hat{\beta}_i(x)$ and train classifier $h$ for one epoch. 3) $\forall i \in [1, K]$, for each sample $(x, \bar{y}, r_x) \in S_i$, update $\hat{\beta}_i(x) = \frac{h_{\text{noisy}}(x)_i}{h(x)_i}$. Then, we repeat steps 2) and 3) through training. In this way, for every epoch, each function $T_{i,i}(\cdot)$ is estimated for the samples from $S_i$. Lastly, for the rest of samples with noisy label $j \neq i$, $T_{i,i}(\cdot)$ is estimated at each epoch using Eq. (4):

$$\forall (x, y, r_x) \in S \backslash S_i, \ \hat{T}_{i,i}(x) = \frac{1}{|S_i|} \sum_{(x', \bar{y}', r'_x) \in S_i} r'_x \hat{\beta}_i(x') = \mu_i. \tag{6}$$

**Computing $\alpha_{i,j}$.** The computation of $\alpha_{i,j}$ boils down to approximating non-diagonal terms of the transition matrix in the CCN model. As $\forall i \neq j, \forall x \in \mathcal{X}, \ T_{i,j}(x) = \alpha_{i,j}(1 - T_{i,i}(x))$, we have:

$$\mathbb{E}_x\left[T_{i,j}(x)\right] = \alpha_{i,j}\left(1 - \mathbb{E}_x\left[T_{i,i}(x)\right]\right) \Leftrightarrow \alpha_{i,j} = \frac{\mathbb{E}_x\left[T_{i,j}(x)\right]}{1 - \mathbb{E}_x\left[T_{i,i}(x)\right]}.$$

A simple and reliable way is to use anchor points, i.e., points for which we can know the true class almost surely. These points may be directly available when some training data has been curated, or they can be identified either theoretically as in Liu & Tao (2015) or heuristically as in Patrini et al. (2017). Having $S_i^* := \{(x, \bar{y}, r_x) \in S | P(Y = i | X = x) \approx 1\}$ a set of class $i$ anchor points, we simply need compute:

$$\forall (x, \bar{y}, r_x) \in S_i^*, \forall j \neq i, \ T_{i,i}(x) = r_x P(\bar{Y} = i | X = x) \ \text{ and } \ T_{i,j}(x) = P(\bar{Y} = j | X = x).$$

Two noisy posteriors can be estimated using the same classifier $h_{\text{noisy}}$ trained on the noisy distribution $h_{\text{noisy}}(x) = \hat{P}(\bar{Y}|X = x)$ aforementioned. Thus, $\alpha_{i,j}$ can be estimated as follows:

$$\forall 1 \leq i, j \leq K, j \neq i, \ \alpha_{i,j} = \frac{\frac{1}{|S_i^*|} \sum_{(x, \bar{y}, r_x) \in S_i^*} h_{\text{noisy}}(x)_j}{1 - \frac{1}{|S_i^*|} \sum_{(x, \bar{y}, r_x) \in S_i^*} r_x h_{\text{noisy}}(x)_i}. \tag{7}$$

**Summary of the training procedure.** Given samples $S$ and $K$ sets of anchor points $(S_i^*)_{i=1}^K$, we want to train a classifier $h(\cdot)$ equipped with a loss $l$. For any loss $l : y, \hat{y} \mapsto l(y, \hat{y})$, we define the $T$-corrected loss as $l_T : y, \hat{y} \mapsto l(y, T\hat{y})$. The overall procedure is in Algorithm 1 (Appendix C).

## 4 EXPERIMENTS

We compare our instance-level forward correction (ILFC) method with four representative baselines: *forward correction* (FC) (Patrini et al., 2017), *mean absolute error* (MAE) (Ghosh et al., 2017), $L_q$-*norm* (LQ) (Zhang & Sabuncu, 2018) and *co-teaching* (CT) (Han et al., 2018). Details are shown in Appendix D. Note that the pioneer IDN methods cannot work for multi-class cases.

### 4.1 SYNTHETIC DATASET

**Generation process.** We generate a synthetic dataset (Appendix E) consisting in three classes of concentric circles (Figure 6a). We then apply the following instance-dependent noise to each label: $P(\bar{Y} \neq Y | X = x) = \rho\left(\frac{w \cdot x}{\|w\|\|x\|} + 1\right)/2$ with $w = (0, 1)$ and $\rho$ controlling the mean noise rate. If corrupted, each label is flipped to another class uniformly.

**Empirical results.** Figure 3 shows the test accuracy of different methods on the synthetic dataset. Each experiment is repeated 5 times and we plot the confidence intervals of each curve. On low-level noise, all methods show good performances (Figure 3a). On mild-level noise, both Co-teaching and ILFC show good performances and outperform other baselines (Figure 3b). On high-level noise, the performance of all the baselines collapse, whereas ILFC constantly maintains good performances (Figures 3c and 3d). More experiments are shown in Appendix B and F.

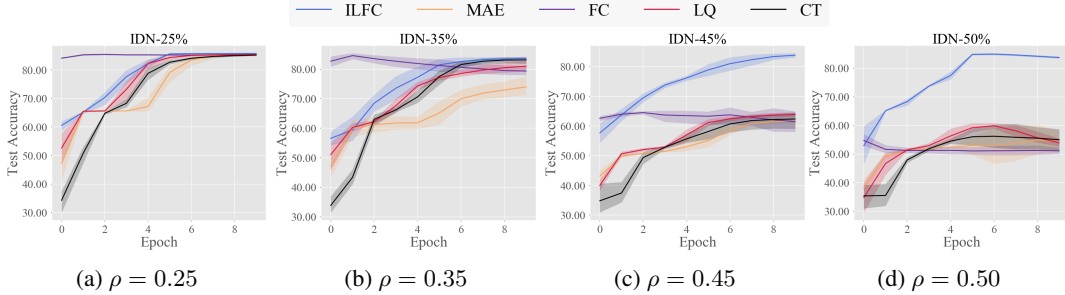

Figure 3: The test accuracy on synthetic datasets with different levels of IDN noise.

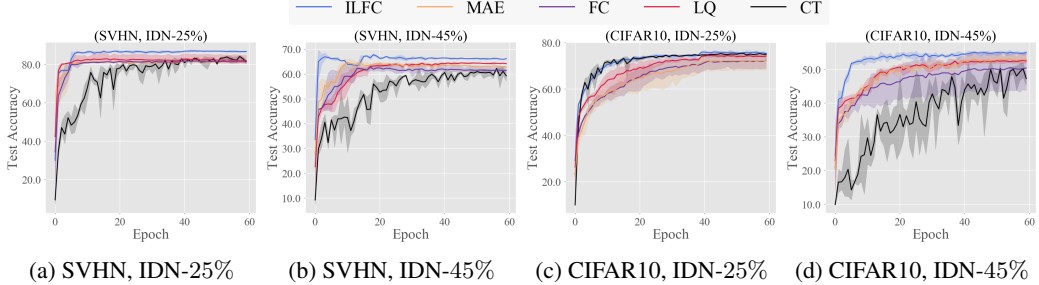

Figure 4: The test accuracy on real-world datasets with different levels of IDN noise.

## 4.2 REAL-WORLD DATASET

**Generation process.** In order to corrupt labels from clean datasets such as SVHN and CIFAR10, we adopt the following procedure: (1) train a classifier $h : x \mapsto \sigma(g(x))$ on a small subset of the clean dataset; (2) using a small validation set, calibrate the classifier by selecting the temperature $t$ that maximizes the expected calibration error as in Guo et al. (2017); (3) for each instance $x$, set: $\bar{y} = \mathrm{argmax}_i \ h_t(x)_i$ and $r_x = \max_i \ h_t(x)_i$. With this process, we attempt to emulate the construction of a real-world dataset (Appendix G).

**Empirical results.** Figures 4a and 4b show the test accuracy on SVHN with $25\%$ and $45\%$ instance-dependent noise, respectively. We can clearly observe that, on both low-level and high-level noise, ILFC shows good performances with a fast convergence rate, and outperforms other baselines. Figures 4c and 4d show the test accuracy on CIFAR10 with $25\%$ and $45\%$ instance-dependent noise, respectively. On low-level noise, all methods show good performances. However, on high-level noise, ILFC shows a fast convergence rate and outperforms other baselines.

## 5 CONCLUSION

In this paper, we give an overview of label-noise learning from class-conditional noise (easier) to instance-dependent noise (harder). We explain why existing approaches cannot handle instance-dependent noise well, and try to address this challenge via confidence scores. Thus, we formally propose the *confidence-scored instance-dependent noise* (CSIDN) model. To tackle the CSIDN model, we design a practical algorithm termed *instance-level forward correction* (ILFC). Our ILFC method robustly outperforms existing methods, especially in the case of high-level noise. In future works, we would like to extend label correction and sample selection approaches with the confidence scores from the CSIDN model.

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

# A   RELATED WORKS

Besides the works aforementioned, we survey other approaches to learning with noisy labels.

**Robust losses.**   Various approaches propose to use a provably robust loss function in the learning process. In the case of class-dependent label noise, Natarajan et al. (2013) constructed an unbiased estimator of any loss function under the noisy distribution. Masnadi-shirazi & Vasconcelos (2009) introduced a robust non-convex loss. Recently, works on symmetric losses showed that such loss offer theoretical robustness results to various types of noise (Ghosh et al., 2017; Charoenphakdee et al., 2019). Motivated by the robustness to noise of the mean absolute error loss (MAE) shown in Ghosh et al. (2017), Zhang & Sabuncu (2018) introduced generalized cross entropy loss that allows for a trade-off between the efficient learning properties of the CCE loss and the noise-robustness of MAE. Shen & Sanghavi (2019) introduced a trimmed loss with an iterative minimization process that allows for theoretical guarantees in the simpler setting of generalized linear models.

**Annotator-level modelling.**   Another recent line of related works attempts to model labels and worker's quality directly during the crowdsourcing annotation process, in order to produce more accurate labels efficiently. Branson et al. (2017) modeled the annotators' skill and instances difficulty while incrementally training a computer vision model during the annotation process, effectively reducing the time burden of the annotation process as well as the error rate in the assigned labels. Guan et al. (2018) modeled each annotator individually in order to better aggregate labels based on each worker's skill and area of expertise. Khetan et al. (2018) introduced a method that allows to learn each workers' skill even when each example is only annotated once, by jointly modelling the assigned labels and the workers during the annotation process.

**Learning with multiple noisy labels.**   A closely related setting is learning from multiple noisy labels, where the aim is to predict an unknown ground-truth label from $(X, (Y^j)_j)$, each $Y^j$ referring to a noisy annotation. This setting can arise for example from crowdsourcing tasks; Snow et al. (2008) showed that using multiple non-expert annotators to train a classifier can be as effective as using gold standard annotations from experts. In Raykar et al. (2009), the authors derive a Bayesian approach to jointly learn the expertise of each annotator, the actual true label and the classifier. Yan et al. (2010) extends this Bayesian approach by considering that each annotator's expertise varies across the input space. This setting differs from ours as it takes place before the aggregation of multiple annotations, which, for CSIDN, is only a way among others to obtain a confidence score for each noisy label.

**Explicit/implicit regularizers.**   Recently, several other regularization techniques have shown good robustness in weakly-supervised settings. Temporal Ensembling (TE) (Laine & Aila, 2017) method labels some additional unlabeled instances using a consensus of predictions from models from previous epochs and with different regularizations and input augmentation conditions. Mean-teacher (MT) (Tarvainen & Valpola, 2017) instead uses predictions from a model obtained by averaging the weights of a set of models similar to TE, as using the prediction from a unique model is more efficient when a large amount of unlabeled data is available. Virtual Adversarial Training (Miyato et al., 2018) regularizes the network using a measure of local smoothness of the conditional label distribution given the input, defined as the robustness of the prediction to local adversarial perturbations in the input space. Introduced in Zhang et al. (2018), *mixup* trains a neural network on convex combinations of instance pairs and their respective labels, and has been shown to reduce the memorization of corrupted labels.

# B   SENSITIVITY ANALYSIS

In practice, the confidence scores obtained may not be accurate. Therefore, we run a sensitivity analysis to assess the robustness of ILFC: similarly to Ishida et al. (2018), we add a zero-mean Gaussian noise with standard deviation $\sigma \in \{0.0, 0.3, 0.6\}$ to each confidence score and clip the values between 0 and 1. Figure 5 shows the resulting performances on the synthetic dataset. ILFC shows good robustness to inaccurate confidence scores even with high standard deviation on a highly noisy dataset.

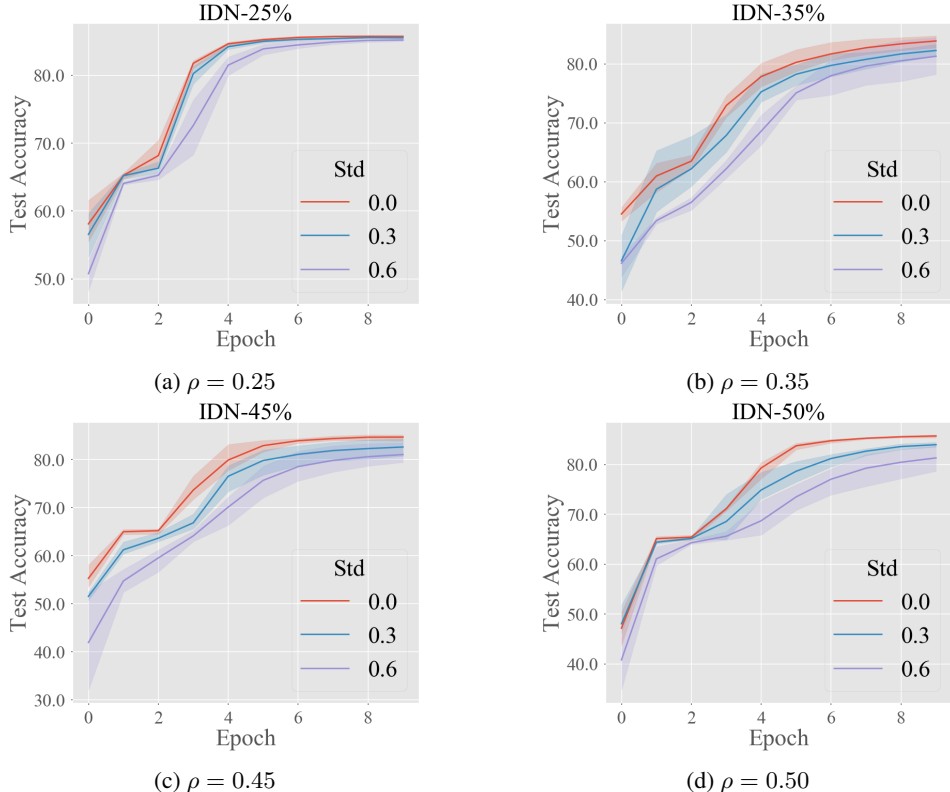

Figure 5: Sensitivity analysis on the synthetic dataset: a zero-mean Gaussian noise of standard deviation $\sigma \in \{0.0, 0.3, 0.6\}$ is added to each confidence score before running ILFC with the noisy confidence scores.

## C  ALGORITHM

We present Algorithm 1 here, which can be referred to Section 3.2 in details.

## D  BASELINES

**Forward correction.**  Introduced in Patrini et al. (2017), *forward correction* estimates a fixed transition matrix $T$ before training, and trains a classifier with the corrected loss $l_T : (y, \hat{y}) \mapsto l(y, T\hat{y})$.

**Mean absolute error loss.**  Due to its symmetric property, the Mean Absolute Error (MAE) has been theoretically justified to be robust to label noise under assumptions (Ghosh et al., 2017). However, this loss is more difficult to train, especially on complex datasets.

$L_q$ **norm.**  Introduced in Zhang & Sabuncu (2018), $L_q$ norm attempts to bring the best of both worlds between the CCE and the MAE loss: the CCE is easy to train, while the MAE is robust to label noise. The authors therefore define this loss using the negative box-cox transformation:

$$L_q\left(h(\boldsymbol{x}), \boldsymbol{e}_j\right) = \frac{(1 - h_j(\boldsymbol{x})^q)}{q},$$

so that the $L_q$ tends to the CCE when $q \to 0$ and to the MAE when $q = 1$. In the following experiments, we set $q = 0.7$, suggested by authors.

**Co-teaching (Han et al., 2018).**  Co-teaching algorithm is a small-loss approach where two classifiers are trained in parallel. At each epoch, each classifier selects the instances with the smallest

---

**Algorithm 1:** Instance-level Forward Correction

**Input** confidence-annotated samples $S := \{(x_i, \bar{y}_i, r_{x_i}), i = 1, \ldots, N\}$, any loss $l$, classifier $h(\cdot)$, anchor points sets $(S_i^*)_{i=1}^K$ ;

(1) Train naive classifier $h_{\text{noisy}}$ on samples $\{(x_i, \bar{y}_i)\}_{i=1}^N$;
(2) $\forall 1 \leq i, j \leq K, i \neq j$, compute $\alpha[i,j]$ from Eq. 7 with anchor points set $S_i^*$;
(3) $\forall 1 \leq i \leq K$, initialize $\beta_i(\cdot) = 1$ ;
**for** *epoch* $N = 1, \ldots, N_{\max}$ **do**

    // Update diagonal constants
    (4) $\forall 1 \leq i \leq K$, compute $\mu[i]$ from Eq. 6;
    **for** $(x, \bar{y}, r_x) \in S$ **do**
        Set $i = \bar{y}$ ;
        // Compute diagonal terms
        (5) Set $T[i,i] = r_x \beta_i(x)$ and $\forall k \in [\![1, K]\!] \backslash \{i\}, T[k,k] = \mu[k]$;
        // Compute non-diagonal terms
        (6) Set $\forall 1 \leq i, j \leq K$, s.t. $i \neq j, T[i,j] = \alpha[i,j](1 - T[i,i])$;
        // Train classifier with instance-level corrected loss
        (7) Train $h(\cdot)$ on sample $(x, \bar{y}, r_x)$ with loss $l_T$;
        // Update density ratio estimate
        (8) Update $\forall 1 \leq i \leq K, \forall x \in S_i, \beta_i(x) = \frac{h_{\text{noisy}_i}(x)}{h_i(x)}$;
    **end**
**end**
(9) Output classifier $h(\cdot)$.

---

loss, and feed them to the other network as a training set for the next iteration. This recent work has proved to be a leading benchmark in the field of noisy labels.

## E   SYNTHETIC DATASET

Figure 6 shows three synthetic datasets, which cover clean, IDN and CSIDN models.

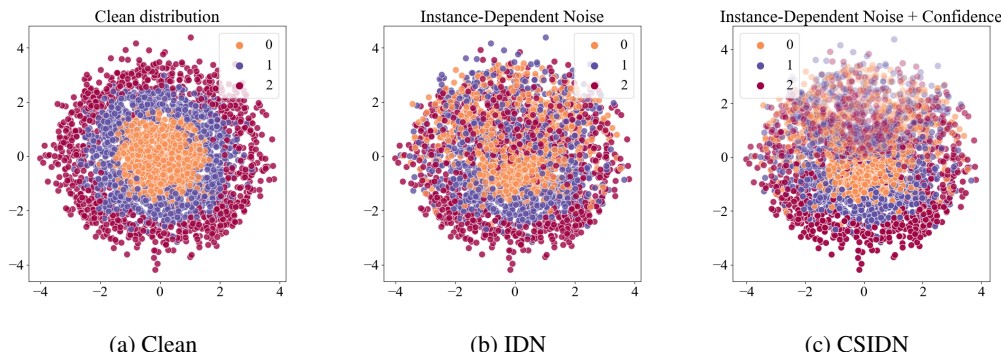

(a) Clean          (b) IDN          (c) CSIDN

Figure 6: Synthetic dataset. The clean distribution (a) consists in three classes of concentric circles. In the IDN setting (b), each point $x$ has a probability $P(\bar{Y} \neq Y | x) = \rho \left( \frac{w \cdot x}{\|w\|\|x\|} + 1 \right) / 2$ with $w = (0, 1)$ of being corrupted, where $\rho$ is a parameter controlling the mean noise rate. Therefore the noise is the strongest towards the direction $(0, 1)$ and the weakest in the direction $(0, -1)$. If corrupted, the label is flipped to another class uniformly. The CSIDN setting (c) is similar to the IDN setting, but each point is associated with measure of the confidence in the assigned label. A lower confidence is represented by a lower opacity in the figure.

## F   DECISION BOUNDARIES

Figure 7 shows the decision boundaries of our approach versus the ones of a benchmark model, for different levels of noise. With high levels of noise, a model that does not include any instance-level

modelling will degenerate around the most noisy region of the input space. On the other hand, our model successfully accounts for the high noise in this region and is able to keep consistent predictions.

## G    EXAMPLES OF REAL-WORLD DATASETS

For example, the method would be similar to constructing a dataset with images scraped from the web, and automatically labelling them from neighbouring text fields using a classifier such as a recurrent neural network. Then, a small subset of curated images could be used at the beginning of the process to calibrate the classifier, in order to make the predictions of the softmax output faithful to the confidence in each label. This way, we could construct a very large dataset for a very low-cost that, while involving some instance-dependent noise, would be equipped with confidence information and therefore could be tackled with our proposed algorithm.

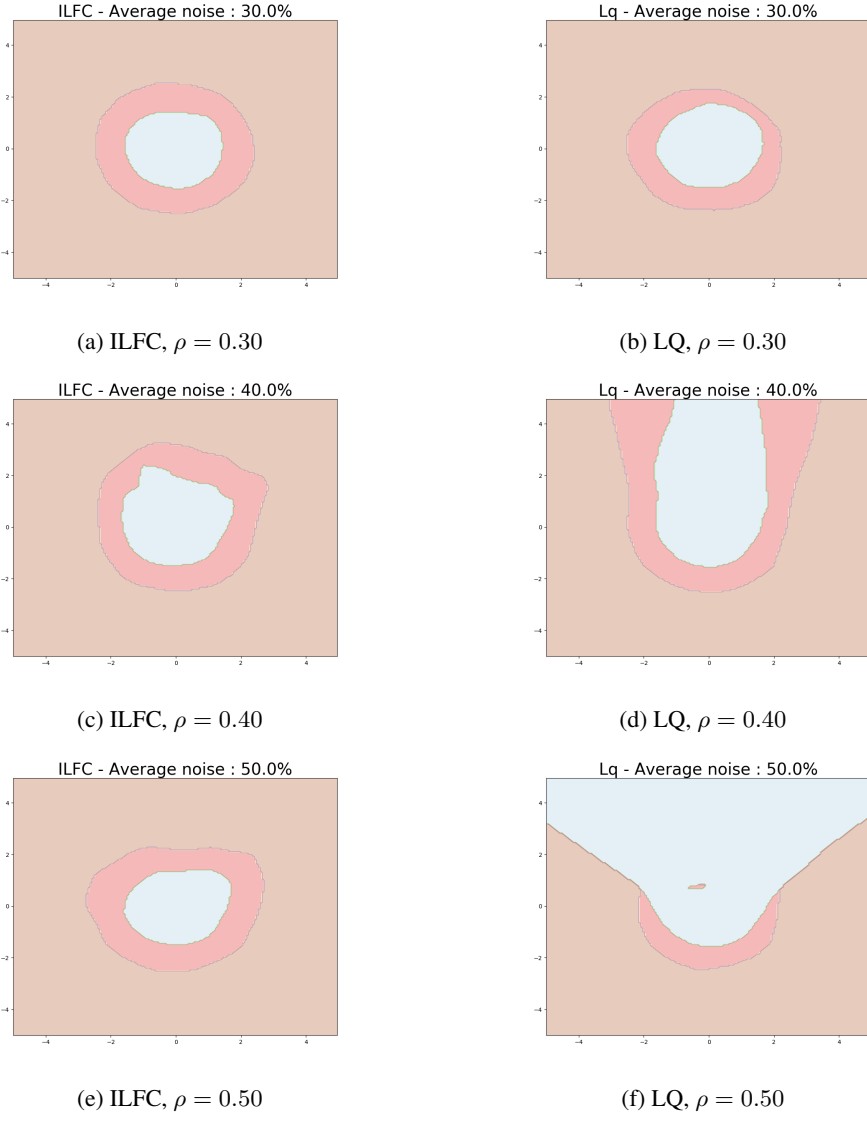

Figure 7: Decision boundaries of the learned classifier for the ILFC model (left column) and the $L_q$ norm model (right column). In the presence of highly noisy regions, a classifier that does not include any instance-level information will degenerate in those regions, while the ILFC approach stays consistent with the clean distribution.

