# OpenReview forum: "Confidence Scores Make Instance-dependent Label-noise Learning Possible"
_ICLR.cc/2020/Conference — Reject_

### Official Review · AnonReviewer1 · 2019-10-22
**Official Blind Review #1**

**Rating:** 8

**Review:**

This paper focuses on instance-dependent label noise problem, which is a new and important area in learning with noisy labels. The authors propose confidence-scored instance-dependent noise (CSIDN) to overcome strong assumptions on noise models. They clearly define confidence scores and justify their availability. To solve CSIDN model, they propose instance-level forward correction with theoretical guarantees. Their experiments on both synthetic and real-world datasets show the advantage of this algorithm.

Pros:

1. This paper is clearly written and well-structured in logic. For example, in Section 2, they introduce from class-conditional noise to instance-dependent noise first, which paves the way for confidence-scored instance-dependent noise. This make readers easy to follow the main contribution, namely the new noise model.

2. This paper pushes the knowledge boundary of learning with noisy labels, since it focuses on more realistic and challenge topic "instance-dependent label noise". The authors leverage the idea of confidence scores, and propose confidence-scored instance-dependent noise (CSIDN). Compared to previous solutions, CSIDN is a tractable instance-dependent noise model, which enjoys several benefits, such as multi-class classification, rate-identifiability and unbound-noise.

3. This paper proposes an algorithm to solve CSIDN inspired by forward correction called instance-level forward correction (ILFC). Their algorithm has been verified in both synthetic datasets and real-world datasets. The empirical results show the advantage of ILFC.

(Minor) cons:

1. Section 3 is a bit dense in understanding the estimation of transition matrix. The authors are encouraged to polish this section.

2. Although ILFC outperform CT and LQ in real-world datasets, the authors need to add the reasults of MAE and FC to more thoroughly verify the performance of ILFC.

**Experience Assessment:**

I have published one or two papers in this area.

**Review Assessment: Checking Correctness Of Derivations And Theory:**

I carefully checked the derivations and theory.

**Review Assessment: Checking Correctness Of Experiments:**

I carefully checked the experiments.

**Review Assessment: Thoroughness In Paper Reading:**

I read the paper thoroughly.

---

> ### Author Response · Authors · 2019-11-07
> **We are running the experiments of FC and MAE on CIFAR10 and SVHN with instance-dependent noise**
>
> Thanks for your appreciation! We are running the experiments of Forward Correction (FC) and MAE on both CIFAR10
> and SVHN with instance-dependent noise. We will update results soon.

---

> > ### Author Response · Authors · 2019-11-11
> > **Empirical results of FC and MAE on CIFAR10 and SVHN with instance-dependent noise**
> >
> > We have added empirical results of FC and MAE on CIFAR10 and SVHN with instance-dependent noise. Please check the updated PDF.
> >
> > We are sorry that Section 3 is a bit dense in understanding the estimation of transition matrix. We will try our best to polish this Section in the next version.

---

> > > ### Comment · AnonReviewer1 · 2019-11-14
> > > **Thank you for your feedback. Score updated**
> > >
> > > Thanks for the prompt reply and new results. I am further convinced and consider the paper strong and worthy of acceptance.

---

### Official Review · AnonReviewer3 · 2019-10-23
**Official Blind Review #3**

**Rating:** 1

**Review:**

--- Overall ---

I think the problem of incorporating information about labeling uncertainty (when such information is available) is an interesting and important problem; however, I think this paper contains a crucial misunderstanding of the definition of calibration (as defined in Guo et al. 2017), is missing some important parts of the literature, and does not provide adequate convergence guarantees for the proposed method.

--- Major comments ---

1. In section 3.1, the authors substitute the confidence score r_x for the conditional probability P(Y=y|\bar{Y}=y,X=x); however, even if the confidence scores are perfectly calibrated, these quantities are not necessarily equal (or even particularly close). As defined in Guo et al. (2017), a confidence score is well calibrated if the probability that a prediction is correct given the confidence score is equal to the confidence score. Using their notation: P(Y=\hat{Y}|\hat{P}=p) = p. Importantly, this is not conditioned on \hat{Y} or X. One way to see that these two quantities are not equal is to observe that there are many possible confidence score functions that satisfy the above definition whereas the conditional probability defined above is a unique function. In particular, if the confidence score function is a constant equal to the accuracy of the model, then the confidence score is well calibrated, but clearly not equal to the conditional probability above.

2. The other scenario considered by the authors is when there are multiple annotators; however, there is substantial literature on learning from multiple noisy annotations which the authors do not review. I suggest the authors start with "Modeling annotator expertise: Learning when everybody knows a bit of something" by Yan et al. (2010)  (and the citations therein) which also considers the instance dependent noise case.

3. Under what conditions does the proposed algorithm converge and to what does it converge to?

4. The proposed method relies on several assumptions that are scattered throughout the description of the algorithm. I highly recommend making these assumptions clear near the beginning of the paper. Specifically, my understanding is that the main assumptions are:

i. Confidence scores r_x are available for each instance and r_x \approx P(Y=y|\bar{Y}=y,X=x).
ii. Y _|_ X | Y\neq\bar{Y}, \bar{Y} = y
iii. Anchor points are available on some portion of the data.

Beyond the presentation, I find this to be a fairly strong set of assumptions, particularly the first assumption.

--- Minor comments ---

1. I really appreciated the synthetic example demonstrating the potential pitfalls of the small loss approach; however, I would spend a bit more time clearly explaining the small loss approach so that readers understand why it fails and how you are solving it's problems.

2. Also in the synthetic example, the authors state that covariate shift leads poor accuracy, however, I think this point would be stronger if demonstrated instead of just asserted.

**Experience Assessment:**

I have published one or two papers in this area.

**Review Assessment: Checking Correctness Of Derivations And Theory:**

I assessed the sensibility of the derivations and theory.

**Review Assessment: Checking Correctness Of Experiments:**

I assessed the sensibility of the experiments.

**Review Assessment: Thoroughness In Paper Reading:**

I read the paper thoroughly.

---

> ### Author Response · Authors · 2019-11-07
> **On the calibration and the fact that r_x is indeed a proper confidence score**
>
> A major concern of R3 is that s/he cannot agree with us r_x = P(Y=y_hat | Y_hat=y_hat, X=x), where Y_hat is the variable representing the predicted label, is a proper confidence score. However, we would like to show that it is a proper confidence score because it can fit the definition of perfect calibration in the calibration area.
>
> The perfect calibration is defined in Guo et al., i.e.,
> P(Y_hat=Y | P_hat=p) = p,
> where P_hat is the confidence score. Note that neither X nor x is involved but this definition is still about the class posterior rather than the class prior. It is because Y_hat implicitly depends on X: Y_hat is a random variable obtained by wrapping a learned classifier f(x) on top of the random variable X, i.e., Y_hat = f(X) and y_hat = f(x). Thus, the random source of Y and Y_hat is from the underlying data distribution (Y itself and X respectively), but the nature of them are completely different.
>
> To make our claim clear, we want to extend the definition of the perfect calibration, i.e.,
> P(Y_hat=Y | P_hat=p_x, X=x) = p_x.
> The extension is reasonable because there is a confidence score for each single data point. In order to make it possible after adding X=x into the condition, we allow p_x to depend on x as well. Observe that X=x makes Y_hat no longer random but just y_hat or f(x); or equivalently, we can regard Y_hat as a one-hot random variable given X=x.
>
> Subsequently, we have
> r_x
> = P(Y=y_hat | Y_hat=y_hat, X=x)
> = P(Y=y_hat, Y_hat=y_hat, X=x) / P(Y_hat=y_hat, X=x)
> = P(Y=y_hat, X=x) / P(X=x)
> = P(Y=y_hat | X=x)
> = P(Y=Y_hat | X=x),
> since Y_hat and y_hat are exactly equivalent given X=x. As a consequence,
> P(Y_hat=Y | r_x=p_x, X=x)
> = P(Y_hat=Y | P(Y=Y_hat|X=x)=p_x, X=x)
> = p_x,
> which concludes our proof that r_x is a proper confidence score similarly to P_hat.
>
> PS, letting the confidence score depend on x is very popular in importance reweighting and related learning methods, for example, a NeurIPS 2018 spotlight paper entitled "binary classification for positive-confidence data".

---

> > ### Comment · AnonReviewer3 · 2019-11-07
> > **RE: Comment 1**
> >
> > Thank you for this clarification! If I understood correctly, what you show in your proof is that if (A) r_x = P(Y=y_hat | Y_hat=y_hat, X=x), then (B) it is a well-calibrated score (i.e. A-->B). I do not disagree with this. What the paper assumes, however, is the converse. That is, the paper assumes that if (B) r_x is well-calibrated, then (A) r_x = P(Y=y_hat | Y_hat=y_hat, X=x) (i.e. B-->A). Fortunately, given the conditional definition of calibration you stated above, this appears true:
> >
> > P(Y=\hat{Y}|\hat{P}=\hat{p}(x), X=x) = \hat{p}(x)
> > P(Y=\hat{y}(x)|\hat{P}=\hat{p}(x), X=x) = \hat{p}(x) (the prediction is deterministic function of X)
> > P(Y=\hat{y}(x)| X=x) = \hat{p}(x) (the confidence score is also a deterministic function of X)
> >
> > where \hat{p}(x) is the determinist mapping of X to a confidence score and \hat{y}(x) is the deterministic mapping of X to a label. However, this raises another problem. Both your proof and mine rely on the assumption that \hat{Y} (and therefore \bar{Y}) is a deterministic function of X. If this is the case, then shouldn't it also be the case that \bar{Y} \perp Y | X? In which case:
> >
> > P(\bar{Y} = i | Y = i, X=x) = P(\bar{Y} = i | X=x) = I[\bar{Y} = \hat{y}(x)]
> >
> > where I[.] is the indicator and \hat{y}(x) is the deterministic mapping from X to \bar{Y}.

---

> > > ### Author Response · Authors · 2019-11-08
> > > **Thanks for your comment and question**
> > >
> > > Yes, it's the case that given X the true label Y and the predicted label Y_hat are statistically independent for any fixed classifier f(x). However, f(x) should be a well trained classifier and it depends on the distributional information of X and Y itself. Thus, even though Y_hat doesn't depend on Y | X, it still depends on p(Y | X) as a whole information source.
> > >
> > > Anyway, it's true that P(Y_hat=i | Y=i, X=x) = I[Y_hat = f(x)] where it doesn't matter which value Y takes. This is because we are considering model evaluation instead of model training, where the predictions don't depend on the true labels but the performance measure does.
> > >
> > > We are sorry for the confusion and will try to clarify this point by simplifying the notations later.

---

> > > > ### Author Response · Authors · 2019-11-11
> > > > **We promise to change the mathematical notations in the next version**
> > > >
> > > > In order to change the mathematical notations, we need to revise the whole paper and carefully proofread it to guarantee the consistent. We promise to do this in the next version.

---

> > > > ### Comment · AnonReviewer3 · 2019-11-14
> > > > **RE: Comment 1**
> > > >
> > > > Thanks for being so responsive and I apologize if I am missing something.
> > > >
> > > > Perhaps I should be more specific: If Y _|_ \bar{Y} | X, then why does equation 3 not reduce to I[\bar{Y} = f(x)] and why does P(Y = i | \bar{Y}=i, X=x) =/= P(Y = i | X=x)?

---

> ### Author Response · Authors · 2019-11-12
> **Thank you for your comments and suggestions**
>
> Thank you for your insightful comments and suggestions. We address
> your remaining comments below.
>
> --- Major comments ---
>
> "2. The other scenario considered by the authors is when there are
> multiple annotators; however, there is substantial literature on
> learning from multiple noisy annotations which the authors do not
> review. I suggest the authors start with "Modeling annotator
> expertise: Learning when everybody knows a bit of something" by Yan et
> al. (2010)  (and the citations therein) which also considers the
> instance dependent noise case."
>
> - Thank you for this reference. We currently give a brief survey of
> works modelling multiple annotator's expertise to produce more
> accurate labels in the Appendix, but we agree that the branch of
> directly learning with multiple noisy annotations is missing from the
> current literature review and we will refer to those works in the following version.
>
>
> "3. Under what conditions does the proposed algorithm converge and to
> what does it converge to?"
>
> - Our method is shown to empirically showed to converge well (cf
> Figures 3 and 4). We plan on studying the proposed algorithm from a
> theoretical viewpoint and derive convergence guarantees in future
> works.
>
>
> "4. The proposed method relies on several assumptions that are
> scattered throughout the description of the algorithm. I highly
> recommend making these assumptions clear near the beginning of the
> paper. Specifically, my understanding is that the main assumptions
> are:
>
> i. Confidence scores r_x are available for each instance and r_x
> \approx P(Y=y|\bar{Y}=y,X=x).
> ii. Y _|_ X | Y\neq\bar{Y}, \bar{Y} = y
> iii. Anchor points are available on some portion of the data.
>
> Beyond the presentation, I find this to be a fairly strong set of
> assumptions, particularly the first assumption."
>
> - These are indeed the main assumptions of our approach. We point out
> that (i) and (ii) are currently stated in the introduction, but we
> agree that (iii) is missing and that the paper would benefit from
> making those three assumptions clearer. We will update the
> introduction accordingly in the following version.
>
>
> --- Minor comments ---
>
> "1. I really appreciated the synthetic example demonstrating the
> potential pitfalls of the small loss approach; however, I would spend
> a bit more time clearly explaining the small loss approach so that
> readers understand why it fails and how you are solving it's problems.
>
> 2. Also in the synthetic example, the authors state that covariate
> shift leads poor accuracy, however, I think this point would be
> stronger if demonstrated instead of just asserted."
>
> Thank you for those additional suggestions. We took due note of them
> and will aim to make those points clearer in the following version.

---

### Official Review · AnonReviewer2 · 2019-10-29
**Official Blind Review #2**

**Rating:** 8

**Review:**

Learning with noise labels is a hot topic now due to the reason that deep learning algorithms often require large-scale supervised training samples and labelling a large amount of data is costly. However, almost all of the existing methods assume that the label noise is instance-independent. It either depends on the clean classes or is completely random. This paper studies the instance-dependent label noise, which is more realistic and applicable, but difficulty to address. The authors target to solve this problem. A feasible solution would contribute to the community a lot.

The challenging part for dealing with instance-dependent label noise is to learn the instance-dependent label flip function, which is hard to learn by only exploiting noisy data without any assumption. Few existing papers proposed assumptions to make the flip function learnable. This paper also introduces an assumption that the confidence score for each data is given. To me, this is also a strong assumption. Although the authors have provided examples of how to collect confidence score, collecting confidence scores may not be easy for many specific problems.

Given the confidence score, the authors proposed to learn the flip function. The clean class posterior can be inferred by using the noisy class posterior and the label flip function. So, how to learn the instance-dependent flip function is an essential step. Some technical contribution has been made to learn the flip function. In this section, the authors further assume that the off-diagonal entries of the flip matrix are independent of instance. After seeing the explanation, I personally agree that the assumption is reasonable for some cases. However, I found that in the experiments, the authors synthesized label noise where the off-diagonal entries depend on the instance. The proposed method also shows its advantages on this case. Can you explain this?

The experiment parts show the effectiveness of the proposed method both on synthetic data and real-world data. Overall, the paper is well-motivated and well-written.

I have another concern that the obtained confidence score may not accurate. Is the proposed method sensitive to the confidence scores?


**Experience Assessment:**

I have read many papers in this area.

**Review Assessment: Checking Correctness Of Derivations And Theory:**

I assessed the sensibility of the derivations and theory.

**Review Assessment: Checking Correctness Of Experiments:**

I carefully checked the experiments.

**Review Assessment: Thoroughness In Paper Reading:**

I read the paper at least twice and used my best judgement in assessing the paper.

---

> ### Author Response · Authors · 2019-11-15
> **Thank you for your appreciation and comments**
>
> Thank you for your appreciation and comments! Please find our responses below.
>
> "This paper also introduces an assumption that the confidence score for each data is given. To me, this is also a strong assumption. Although the authors have provided examples of how to collect confidence score, collecting confidence scores may not be easy for many specific problems."
>
> - We agree that this is a substantial assumption. However, without any additional assumption the IDN setting is an ill-posed problem, hence the current lack of feasible solutions for real-world applications. With this assumption, we aim to propose a possible solution to this setting that has real-world applications as shown by the proposed examples.
>
>
> "In this section, the authors further assume that the off-diagonal entries of the flip matrix are independent of instance. After seeing the explanation, I personally agree that the assumption is reasonable for some cases . However, I found that in the experiments, the authors synthesized label noise where the off-diagonal entries depend on the instance. The proposed method also shows its advantages on this case. Can you explain this?"
>
> - Please note that we only assume that the off-diagonal entries of the flip matrix are independent of the instance conditionally on (\hat{Y} \neq Y). Therefore, following Eq. (5), we still have T_ij, i \neq j as a function of x but T_ij(x) is only dependent of x through the diagonal term T_ii(x) since our assumptions boils down to \alpha_ij(x) = \alpha_ij. Sorry for the confusion, we will clarify this point in the following version.
>
>
> "I have another concern that the obtained confidence score may not accurate. Is the proposed method sensitive to the confidence scores?"
>
> - We are currently running some sensitivity analysis on the confidence scores and we will add the results in the following version.

---

### Decision · Program_Chairs · 2019-12-19

**Decision:**

Reject

**Comment:**

While two reviewers  rated this paper as an accept, reviewer 3 strongly believes there are unresolved issues with the work as summarized in their post-rebuttal review. This work seems very promising and while the AC will recommend rejection at this time, the authors are strongly encouraged to resubmit this work.